# Proteome-wide Mendelian randomization identifies causal links between blood proteins and severe COVID-19

Alish B. Palmos[1,2‡], Vincent Millischer[3,4‡]*, David K. Menon[5], Timothy R. Nicholson[2,6], Leonie S. Taams[7], Benedict Michael[8], Geraint Sunderland[9,10], Michael J. Griffiths[9,11], COVID Clinical Neuroscience Study Consortium, Christopher Hübel[1,2,12‡], Gerome Breen[1,2‡]

1 Social, Genetic & Developmental Psychiatry Centre, Institute of Psychiatry, Psychology & Neuroscience, King's College London, London, United Kingdom, 2 UK National Institute for Health Research (NIHR) Biomedical Research Centre for Mental Health, South London and Maudsley Hospital, London, United Kingdom, 3 Department of Psychiatry and Psychotherapy, Medical University of Vienna, Vienna, Austria, 4 Department of Molecular Medicine and Surgery, Karolinska Institutet, Stockholm, Sweden, 5 Division of Anaesthesia, Department of Medicine, University of Cambridge, Cambridge, United Kingdom, 6 Institute of Psychiatry, Psychology & Neuroscience, King's College London, London, United Kingdom, 7 Centre for Inflammation Biology & Cancer Immunology, Department of Inflammation Biology, School of Immunology & Microbial Sciences, King's College London, London, United Kingdom, 8 Institute of Infection and Global Health, University of Liverpool, Liverpool, United Kingdom, 9 Department of Clinical Infection, Microbiology and Immunology, Institute of Infection, Veterinary and Ecological Sciences, University of Liverpool, Liverpool, United Kingdom, 10 Department of Neurosurgery, Alder Hey Children's NHS Trust, Liverpool, United Kingdom, 11 Department of Neurology, Alder Hey Children's NHS Trust, Liverpool, United Kingdom, 12 National Centre for Register-based Research, Department of Economics and Business Economics, Aarhus University, Aarhus, Denmark

‡ ABP and VM indicates joint first authorship on this work. CH and GB are joint last authorship on this work.
* vincent.millischer@meduniwien.ac.at

**Data Availability Statement:** The analyses are based on open data. The links and origin of the summary statistics have been added as a column in S1 Table: https://www.ebi.ac.uk/gwas/

## Abstract

In November 2021, the COVID-19 pandemic death toll surpassed five million individuals. We applied Mendelian randomization including >3,000 blood proteins as exposures to identify potential biomarkers that may indicate risk for hospitalization or need for respiratory support or death due to COVID-19, respectively. After multiple testing correction, using genetic instruments and under the assumptions of Mendelian Randomization, our results were consistent with higher blood levels of five proteins GCNT4, CD207, RAB14, C1GALT1C1, and ABO being causally associated with an increased risk of hospitalization or respiratory support/death due to COVID-19 (ORs = 1.12–1.35). Higher levels of FAAH2 were solely associated with an increased risk of hospitalization (OR = 1.19). On the contrary, higher levels of SELL, SELE, and PECAM-1 decrease risk of hospitalization or need for respiratory support/death (ORs = 0.80–0.91). Higher levels of LCTL, SFTPD, KEL, and ATP2A3 were solely associated with a decreased risk of hospitalization (ORs = 0.86–0.93), whilst higher levels of ICAM-1 were solely associated with a decreased risk of respiratory support/death of COVID-19 (OR = 0.84). Our findings implicate blood group markers and binding proteins in both hospitalization and need for respiratory support/death. They, additionally, suggest that higher levels of endocannabinoid enzymes may increase the risk of hospitalization. Our research replicates findings of blood markers previously associated with COVID-19 and

publications/23696881 https://www.ebi.ac.uk/gwas/publications/28240269 https://www.ebi.ac.uk/gwas/publications/27989323 https://www.ebi.ac.uk/gwas/publications/28369058 https://www.ebi.ac.uk/gwas/publications/29875488 https://www.ebi.ac.uk/gwas/publications/31217265 (Contacted corresponding author for full summary statistics). https://datashare.ed.ac.uk/handle/10283/3649 https://zenodo.org/record/2615265#.YaD2uJDMLOQ https://www.ebi.ac.uk/gwas/publications/31727947 https://www.ebi.ac.uk/gwas/publications/31320639 All results can be found in S1_Data.xlsx Analyses scripts are publicly available under https://github.com/tnggroup/PWMR_Covid19.

**Funding:** BM and GB are supported to conduct COVID-19 neuroscience research (The Covid-19 Clinical Neuroscience Study (COVID-CNS)) by the Medical Research Council (UKRI/MRC; MR/V03605X/1); for additional neurological inflammation research due to viral infection BM is also supported by grants from the MRC/UKRI (MR/V007181//1), MRC (MR/T028750/1) and Wellcome (ISSF201902/3). CH acknowledges funding from Lundbeckfonden (R276-2018-4581). MJG is supported for neuroscience research internationally by MRC Newton Fund (MR/S019960/1), MRC Developmental Pathway Funding Scheme (MR/R015406/1), and National Institute for Health Research (NIHR; 153195 17/60/67, 126156 17/63/11, and 200907). DKM is also funded by the NIHR (through the Cambridge NIHR Biomedical Research Centre) and by the Addenbrooke's Charities Trust. This paper represents independent research partially funded by the National Institute for Health Research (NIHR) Biomedical Research Centre (BRC) at the South London and Maudsley NHS Foundation Trust and King's College London. The authors acknowledge use of the research computing facility at King's College London, Rosalind (https://rosalind.kcl.ac.uk), which is delivered in partnership with the National Institute for Health Research (NIHR) Biomedical Research Centres at South London & Maudsley and Guy's & St. Thomas' NHS Foundation Trusts, and part-funded by capital equipment grants from the Maudsley Charity (award 980) and Guy's & St. Thomas' Charity (TR130505). The funders had no role in study design, data collection, and analysis, decision to publish, or preparation of the manuscript.

**Competing interests:** I have read the journal's policy and the authors of this manuscript have the following competing interests: Gerome Breen has sat on pre-clinical advisory boards for Compass

prioritises additional blood markers for risk prediction of severe forms of COVID-19. Furthermore, we pinpoint druggable targets potentially implicated in disease pathology.

## Author summary

As of November 2021, more than five million people have died due to COVID-19. Although vaccinations provide good protection, it is important to fully understand the biology behind the severe forms of COVID-19. Mendelian randomization facilitates the identification of blood proteins that may be involved in the pathophysiology of severe forms. Here, we investigated whether >3,000 blood proteins might play a role in hospitalization due to COVID-19 or the requirement of respiratory support or death due to COVID-19. Using genetic instruments and under the assumption of Mendelian randomization, our results are consistent with higher levels of five proteins being causally associated with an increased risk of both COVID-19 outcomes and higher levels of one protein associated with hospitalization. Our results are also consistent with higher levels of four proteins–mainly playing a role in cell adhesion–being causally associated with a decreased risk of hospitalization and respiratory support/death, and higher levels of four proteins being causally associated with a decreased risk of hospitalization. These proteins may represent new biomarkers useful in risk prediction of severity and may lead to new therapeutics by prioritizing druggable targets.

## Introduction

The severe acute respiratory syndrome coronavirus 2 (SARS-CoV-2) was identified in late 2019 in Wuhan, China, and is commonly referred to as coronavirus disease 2019 (COVID-19) that rapidly evolved into a global pandemic [1,2]. As of November 2021, more than 240 million cases have been confirmed worldwide with total deaths exceeding 5 million [3]. COVID-19 pathology encompasses a wide spectrum of clinical manifestations from asymptomatic, mild, moderate, to 15% being severe infections [1,2,4]. Severe COVID-19 commonly requires hospitalization and intensive care with assisted respiratory support, and respiratory failure is the most common reason for COVID-19 associated mortality [5].

A dysregulated pro- and anti-inflammatory immunomodulatory response is thought to drive much of the pathophysiology of COVID-19 and comprises alveolar damage, lung inflammation and pathology of an acute respiratory distress syndrome [1,2,6,7]. Given that the innate immune response has an individual-level genetic basis, genetic variants carried by an individual could play an important role in the individual-level immune response and, therefore, may influence progression and severity of COVID-19. This individual difference may also be key in our understanding of why some individuals require hospitalization due to the severity of their symptoms, whilst others are able to recover from COVID-19 without hospitalization [8]. In addition, once hospitalized, this individual difference may drive some people towards fatal outcomes or intensive care with respiratory support, whilst others are discharged from hospitals without respiratory complications.

Although immunomodulatory blood proteins can be studied in hospitalized and non-hospitalized patients with COVID-19, it is difficult to avoid potential confounding effects through factors, such as initial viral exposure/inoculum, smoking behavior, and high body mass index (BMI). These factors themselves are associated with high pro-inflammatory cytokine levels

Pathways Ltd and Otsuka Ltd. Michael J Griffiths
has received research supported by Fast Track
Diagnostics, Luxembourg and consultancy fees
from Siemens Healthineers. There are no other
conflicts to disclose.

and may represent independent risk factors for hospitalization, death, or respiratory failure as a result of COVID-19 [9,10].

Numerous genome-wide association studies (GWASs) in healthy populations associated genetic variants with immunomodulatory blood proteins [11–13]. In addition, the COVID-19 Host Genetics Initiative carried out GWASs on COVID-19 outcomes to understand the role of host genetic factors in susceptibility and severity of COVID-19 [8]: the first GWAS associated genetic variants with hospitalization due to COVID-19; and the second with need for respiratory support and death subsequently to a COVID-19 hospitalization. The findings suggest that once hospitalized another set of genetic variants may be responsible for a severe respiratory form of COVID-19, which may lead to the need for respiratory support or death.

These GWASs represent a powerful source of information to identify new biomarkers and therapeutic leads for drug development or repositioning. The method of Mendelian randomization can investigate the relationship between immunomodulatory blood proteins and a severe COVID-19 infection. Mendelian randomization exploits the fact that alleles are randomly inherited from parent to offspring in a manner analogous to a randomized-controlled trial, and allows estimation of putative causal effects of an exposure on a disease while avoiding confounding environmental effects, thus overcoming some of the limitations of observational studies. Recent advancements in Mendelian randomization methods allow use of GWAS summary statistics to identify genetic proxies (i.e., instrumental variables) of modifiable risk factors and test their association with disease outcomes [14,15]. We, therefore, conducted Mendelian randomization analyses between high levels of a large number of blood proteins and COVID-19, highlighting specific proteins associated with an increased risk of hospitalization due to COVID-19 and once hospitalized, an increased risk for need of respiratory support/death due to COVID-19. We identified putative causal associations that help us understand how innate differences in protein levels can affect the COVID-19 disease course and which proteins could be prioritized in clinical studies.

## Methods

### Blood protein GWAS data

In total, we amassed 5,305 sets of GWAS summary statistics for blood biomarkers [11–13,16–23]. A systematic search was performed based on the ontology lookup service (OLS; www.ebi.ac.uk/ols/index) using R and the packages 'rols' and 'gwasrapidd' between July 7th and July 27th, 2020. OLS is a repository for biomedical ontologies, such as gene ontology (GO) or the experimental factor ontology (EFO), including a systematic description of many experimental variables. First, all subnodes of the EFO 'protein measurements' (EFO:0004747) were determined using an iterative process based on the package 'rols'. Overall, 628 unique EFO IDs were determined. Subsequently, all genetic associations reported in the GWAS Catalog [24] (www.ebi.ac.uk/gwas/) were identified and linked to the corresponding study using the package 'gwasrapidd'. One hundred and seventy-eight unique GWAS catalog accession IDs with available summary statistics were curated manually before inclusion. In order to expand the dataset, studies published at a later date were included manually and the first and the last author of studies without publicly available summary statistics were contacted.

In total, we included ten publications for which summary data was readily available and processed those using standard GWAS summary statistics quality control metrics including removal of incomplete genetic variants, variants with information metrics of lower than 0.6 and allele frequencies more extreme than 0.005 or 0.995. Allele frequencies were estimated from raw genotypes of the European 1,000 Genomes Project dataset, where needed [25]. See **S1 Table** for a full list of studies included in these analyses. Links to the summary statistics are

also provided. Note that all protein measurements from all studies described above were included in our analyses, meaning that some proteins were analyzed more than once.

## COVID-19 GWAS data

In order to capture increased risk of hospitalization as a result of severe COVID-19, we downloaded the COVID-19 Host Genome Initiative GWAS meta-analysis [8] of "Hospitalized covid vs. population", European ancestry (B2_ALL_eur, release 5, January 2021; https://www. covid19hg.org/results/r5/). Cases were defined as SARS-CoV-2 infected individuals who required hospitalization due to COVID-19 related symptoms. Controls were defined as non-cases, i.e. the population. The European sample consisted of 9,986 cases and 1,877,672 controls. In our study, we refer to this GWAS as the hospitalization-COVID-19 GWAS.

In order to capture increased risk of very severe respiratory COVID-19, including respiratory support and death, we downloaded the COVID-19 Host Genome Initiative GWAS meta-analysis [8] of "very severe respiratory confirmed covid vs. population", European ancestry (A2_ALL_eur, release 5, January 2021; www.covid19hg.org/results/r5). Cases were defined as SARS-CoV-2 infected individuals who were admitted to hospital, had COVID-19 as the primary reason for admission, and had died or needed respiratory support (i.e., intubation, continuous positive airway pressure, or bilevel positive airway pressure). Controls were defined as non-cases, i.e. the population [8]. The European sample consisted of 5,101 cases and 1,383,241 controls. In this study, we refer to this GWAS as the respiratory support/death-COVID-19 GWAS.

## Mendelian randomization

To examine the influence of blood proteins on the risk of developing severe COVID-19, we selected genetic variants, single nucleotide polymorphisms (SNPs), that were strongly associated with actual blood protein levels in 5,504 genome-wide analyses of single proteins using robust methodologies (see **S1 Data**, for more details on how the proteins were measured, and instruments for all significant proteins). Using these genetic loci as proxies for protein levels, we performed an analysis using Mendelian randomization, a method that enables tests of putative causal associations of these blood proteins with the development of severe COVID-19. We used the Generalized Summary data-based Mendelian randomization (GSMR) method as the base method [26]. GSMR tests for putative causal associations between a risk factor and a disease using multi-SNP effects from GWAS summary data. The HEIDI-outlier approach in GSMR removes SNP instruments with strong putative pleiotropic effects. In addition, GSMR accounts for linkage disequilibrium (LD) among SNPs not removed by clumping using a reference dataset for LD estimation. In this study, the European 1,000 Genomes dataset was used as the reference dataset [18].

For all GWASs, SNPs used as instrumental variables were selected by applying a suggestive genome-wide *p-value* threshold ($p < 5 \times 10^{-6}$), to identify enough SNPs (i.e., at least 5) in common between the exposure (e.g., blood marker) and outcome (e.g. COVID-19 hospitalization). Note that although reducing the *p-value* threshold may introduce potential false positive SNPs as instruments, SNPs with the strongest effect sizes are robust and reliable for conducting MR. The use of a lower *p-value* threshold in numerous MR studies is common [27–31], and we additionally calculated F-statistics and I-squared statistics to transparently present the strength of our instruments (**S2 Table**). We needed to take this analytical step as GWAS of blood proteins with more statistical power are not available at this time.

Where possible (due to SNPs in common between the exposure and the outcome), bidirectional analyses were performed. To account for multiple testing, we calculated false discovery rate (FDR) corrected $Q$ values using the p.adjust function in R ($p_{FDR} = 0.05$) [32].

## Sensitivity analyses

To test for robustness, we performed sensitivity analyses on our significant results from the GSMR analyses using additional Mendelian randomization methods, including the maximum likelihood, MR Egger, simple median, weighted median, inverse weighted median, inverse weighted median radial, inverse variance weighted (multiplicative random effects), inverse variance weighted (fixed effects), simple mode and weighted mode methods [33–35]. In order to pass our sensitivity analyses, at least nine of these ten methods must agree with the primary GSMR results.

Furthermore, when possible, we performed GSMR using only variants in the cis region of the gene encoding the blood marker (defined as variants either within a gene, up to 1 Mb proximal to the start of the gene, or up to 1 Mb distal to the end of the gene). Gene information was obtained from ensembl [36] using the biomaRt library [37], SNP information was obtained from NCBI dbSNP [38] using the rsnps library [39] (Gustavsen et al., "Get 'SNP' ('Single-Nucleotide' 'Polymorphism') Data on the Web [R Package Rsnps Version 0.4.0]" 2020).

With BMI being associated with both COVID-19 severity and the levels of many inflammatory proteins [9,40,41], we also performed GSMR with BMI both as exposure and outcome for all significant proteins.

Finally, given that many inflammatory and immunomodulatory proteins share genetic loci and may therefore be driving associations via genetically correlated SNPs in high linkage disequilibrium, we computed pairwise linkage disequilibrium for all SNPs used as instrumental variables of blood proteins that were significantly associated in our analyses. To calculate linkage disequilibrium, we used LDlink [42] and the CEU population panel (Utah residents from North and West Europe) as the reference.

## Pathway analyses

KEGG pathway analysis was performed in R with the significant proteins for both outcomes separately, using the clusterProfiler library 4.0 [43]. To account for multiple testing, we calculated false discovery rate (FDR) on the pathways, the significance threshold was set at $p_{FDR} = 0.05$.

## Results

We tested 3,890 associations with hospitalization-COVID-19 as the exposure and blood proteins as outcome (yielding 1 statistically significant association); and in reverse, 5,314 associations of blood proteins as the exposure and hospitalization-COVID-19 as the outcome (yielding 15 statistically significant associations). Additionally, we tested 2,687 associations with need for respiratory support/death-COVID-19 as the exposure (yielding 1 significant association); and in reverse, 3,273 associations with respiratory support/death-COVID-19 as the outcome (yielding 13 significant associations, **Table 1**). Our results show for some proteins, robust associations with the same proteins twice, as they were measured twice in independent GWASs, serving as direct replication. In order to easily identify these proteins, we added a suffix of the study name to the protein. In addition, note that units of protein measurement differed in studies, with some studies using standardized units (see studies in **S1 Table**). Thus, we will report our findings per standard deviation increase.

**Table 1. Mendelian randomization results with COVID-19.** Details of significant, false discovery rate-(FDR)-corrected Mendelian randomization results, using the Generalized Summary Data-based Mendelian randomization (GSMR) method. Using genetic instruments and under the assumptions of Mendelian randomization, the top section shows results consistent with six blood markers being significantly causally associated with an increased risk of hospitalization as a result of COVID-19 and the nine blood markers causally associated with a decreased risk of hospitalization, as well as one protein showing a decrease in risk of hospitalization. The bottom section shows results consistent with five blood markers being significantly causally associated with an increased risk for the need of respiratory support/death due to COVID-19 and eight blood markers causally associated with a decreased risk for the need of respiratory support/death due to COVID-19, as well as one protein decreasing risk for the need of respiratory support/death due to COVID-19. The table presents the log odds statistics (i.e., beta) and corresponding standard error as well as odds ratios, 95% confidence intervals, and the FDR-adjusted Q values ($p_{FDR} = 0.05$).

| Outcome—Hospitalization as a Result of COVID-19 | | | | | | | | |
|---|---|---|---|---|---|---|---|---|
| Protein (exposure) | Beta | SE | p value | SNPs | OR | Lower 95% CI | Upper 95% CI | Q |
| FAAH2_Sun | 0.17 | 0.03 | $1.54 \times 10^{-07}$ | 13 | 1.19 | 1.12 | 1.25 | $1.31 \times 10^{-04}$ |
| GCNT4_Sun | 0.15 | 0.03 | $8.15 \times 10^{-08}$ | 18 | 1.16 | 1.11 | 1.21 | $8.68 \times 10^{-05}$ |
| CD207_Sun | 0.11 | 0.02 | $1.95 \times 10^{-08}$ | 24 | 1.11 | 1.08 | 1.15 | $3.59 \times 10^{-05}$ |
| RAB14_Sun | 0.10 | 0.02 | $3.77 \times 10^{-08}$ | 24 | 1.11 | 1.07 | 1.14 | $5.29 \times 10^{-05}$ |
| C1GALT1C1_Sun | 0.10 | 0.02 | $4.06 \times 10^{-05}$ | 26 | 1.10 | 1.06 | 1.15 | $2.47 \times 10^{-02}$ |
| ABO_Sun | 0.07 | 0.01 | $1.36 \times 10^{-07}$ | 29 | 1.07 | 1.05 | 1.10 | $1.29 \times 10^{-04}$ |
| LCTL_Sun | -0.08 | 0.02 | $1.51 \times 10^{-06}$ | 40 | 0.93 | 0.90 | 0.96 | $1.07 \times 10^{-03}$ |
| SFTPD_Breth | -0.08 | 0.02 | $6.36 \times 10^{-05}$ | 16 | 0.92 | 0.88 | 0.96 | $3.61 \times 10^{-02}$ |
| SELL_Sun | -0.09 | 0.02 | $3.65 \times 10^{-07}$ | 24 | 0.91 | 0.88 | 0.95 | $2.83 \times 10^{-04}$ |
| SELE_Folk | -0.11 | 0.02 | $4.35 \times 10^{-08}$ | 16 | 0.90 | 0.86 | 0.94 | $5.29 \times 10^{-05}$ |
| KEL_Sun | -0.11 | 0.03 | $9.05 \times 10^{-05}$ | 18 | 0.90 | 0.84 | 0.95 | $4.54 \times 10^{-02}$ |
| SELE_Scal | -0.12 | 0.02 | $2.11 \times 10^{-08}$ | 50 | 0.88 | 0.84 | 0.93 | $3.59 \times 10^{-05}$ |
| SELE_Breth | -0.13 | 0.03 | $8.49 \times 10^{-06}$ | 6 | 0.88 | 0.82 | 0.94 | $5.57 \times 10^{-03}$ |
| ATP2A3_Sun | -0.16 | 0.04 | $8.99 \times 10^{-05}$ | 16 | 0.86 | 0.78 | 0.93 | $4.54 \times 10^{-02}$ |
| PECAM1_Scal | -0.23 | 0.04 | $2.05 \times 10^{-10}$ | 30 | 0.80 | 0.73 | 0.87 | $1.74 \times 10^{-06}$ |
| Exposure—Hospitalization as a Result of COVID-19 | | | | | | | | |
| Protein (outcome) | Beta | SE | p value | SNPs | | | | Q |
| MIP1b_Ahol | -0.16 | 0.03 | $7.76 \times 10^{-09}$ | 27 | | | | $2.26 \times 10^{-05}$ |
| Outcome—Respiratory support/death due to COVID-19 | | | | | | | | |
| Protein (exposure) | Beta | SE | p value | SNPs | OR | Lower 95% CI | Upper 95% CI | Q |
| GCNT4_Sun | 0.30 | 0.05 | $3.36 \times 10^{-11}$ | 16 | 1.35 | 1.26 | 1.44 | $6.68 \times 10^{-08}$ |
| RAB14_Sun | 0.20 | 0.03 | $1.32 \times 10^{-11}$ | 27 | 1.22 | 1.16 | 1.28 | $3.92 \times 10^{-08}$ |
| C1GALT1C1_Sun | 0.19 | 0.04 | $3.19 \times 10^{-07}$ | 28 | 1.21 | 1.13 | 1.28 | $2.37 \times 10^{-04}$ |
| CD207_Sun | 0.16 | 0.03 | $2.97 \times 10^{-07}$ | 25 | 1.17 | 1.11 | 1.23 | $2.37 \times 10^{-04}$ |
| ABO_Sun | 0.11 | 0.02 | $8.35 \times 10^{-08}$ | 30 | 1.12 | 1.08 | 1.16 | $8.30 \times 10^{-05}$ |
| SELE_Sliz | -0.11 | 0.02 | $5.87 \times 10^{-06}$ | 65 | 0.89 | 0.84 | 0.94 | $3.18 \times 10^{-03}$ |
| SELL_Sun | -0.13 | 0.03 | $9.03 \times 10^{-06}$ | 24 | 0.88 | 0.83 | 0.94 | $4.49 \times 10^{-03}$ |
| SELE_Scal | -0.17 | 0.04 | $4.35 \times 10^{-06}$ | 52 | 0.85 | 0.78 | 0.92 | $2.59 \times 10^{-03}$ |
| sICAM1_Sliz | -0.17 | 0.04 | $2.98 \times 10^{-05}$ | 31 | 0.84 | 0.76 | 0.92 | $1.27 \times 10^{-02}$ |
| SELE_Folk | -0.19 | 0.03 | $9.44 \times 10^{-10}$ | 16 | 0.83 | 0.77 | 0.89 | $1.41 \times 10^{-06}$ |
| SELE_Breth | -0.20 | 0.05 | $1.87 \times 10^{-05}$ | 6 | 0.82 | 0.73 | 0.91 | $8.57 \times 10^{-03}$ |
| PECAM1_Folk | -0.26 | 0.05 | $1.49 \times 10^{-06}$ | 8 | 0.77 | 0.66 | 0.88 | $9.87 \times 10^{-04}$ |
| PECAM1_Scal | -0.31 | 0.05 | $1.34 \times 10^{-09}$ | 31 | 0.73 | 0.63 | 0.83 | $1.60 \times 10^{-06}$ |
| Exposure—Respiratory support/death due to COVID-19 | | | | | | | | |
| Protein (outcome) | Beta | SE | p value | SNPs | | | | Q |
| NEP_Hill | -0.28 | 0.07 | $7.63 \times 10^{-05}$ | 24 | | | | $3.03 \times 10^{-02}$ |

Note: SE = standard error, SNPs = number of single nucleotide polymorphisms in common between the exposure and outcome; OR = odds ratio, CI = confidence interval, Q = false discovery rate-adjusted p value; ABO = ABO system transferase; ATP2A3 = ATPase Sarcoplasmic/Endoplasmic Reticulum Ca2+ Transporting 3; C1GALT1C1 = C1GALT1 specific chaperone 1; CD207 = langerin; FAAH2 = Fatty Acid Amide Hydrolase 2; GCNT4 = glucosaminyl (N-Acetyl) transferase 4; KEL = Kell Metallo-Endopeptidase (Kell Blood Group); LCTL = Lactase-like protein; MIP1b = macrophage inflammatory protein; NEP = neprilysin; PECAM1 = platelet endothelial cell adhesion molecule; RAB14 = ras-related protein rab-14; SELE = E-selectin; SELL = L-selectin; SFTPD = Surfactant Protein D; sICAM1 = Soluble intercellular adhesion molecule-1. Names after the underscore are abbreviations for the study the protein was measured in.

## Proteins associated with an elevated risk of hospitalization as a result of COVID-19

After multiple testing correction ($p_{FDR}$ = 0.05), using genetic instruments and under the assumptions of Mendelian randomization, our results were consistent with six blood markers being significantly causally associated with an elevated risk of hospitalization as a result of COVID-19 (**Figs 1 and S1**); the reverse associations with risk of hospitalization as exposure and these six blood markers as outcome revealed no significant associations (**S3 Table**). Per

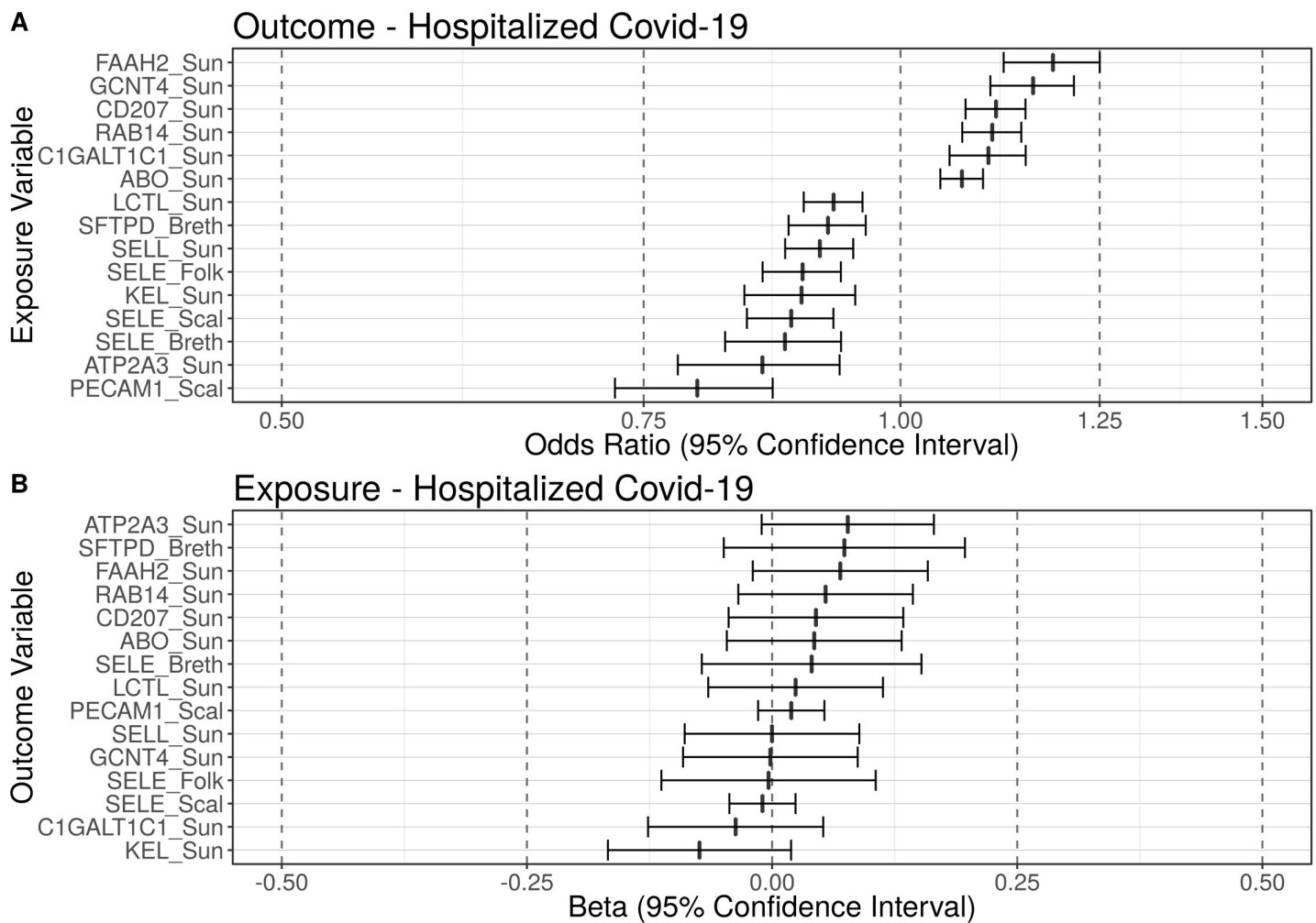

**Fig 1. Blood markers putatively causally associated with hospitalized COVID-19.** Summary figure of the false discovery rate-corrected ($p_{FDR}$ = 0.05) Mendelian randomization results using the Generalized Summary data-based Mendelian randomization (GSMR) method. Using genetic instruments and under the assumptions of Mendelian randomization, this figure displays: (A) Summary figure when hospitalization-COVID-19 is the outcome of interest; (B) Summary figure when hospitalization-COVID-19 is the exposure of interest. Odds ratios (ORs) of the blood markers causally associated with hospitalized-Covid-19 are displayed on the x-axis (with 95% confidence intervals). The blood markers are displayed on the y-axis. The dashed line at one represents an odds ratio of one (i.e., no effect). Using genetic instruments and under the assumptions of Mendelian randomization, six blood markers were causally associated with a significantly increased risk for hospitalization COVID-19 and nine blood markers were causally associated with a significantly decreased risk for hospitalization ($q_{FDR} \leq 0.05$). ABO = ABO system transferase; ATP2A3 = ATPase sarcoplasmic/endoplasmic reticulum Ca2+ transporting 3; C1GALT1C1 = C1GALT1 specific chaperone 1; CD207 = langerin; FAAH2 = fatty acid amide hydrolase 2; GCNT4 = glucosaminyl (N-Acetyl) transferase 4; KEL = Kell metallo-endopeptidase (Kell Blood Group); LCTL = lactase-like protein; PECAM1 = platelet endothelial cell adhesion molecule; RAB14 = ras-related protein rab-14; SELE = E-selectin; SELL = L-selectin; SFTPD = surfactant protein D.

standard deviation (SD) increase in the respective blood marker our results were consistent with an increase in odds for hospitalization ranging from 7 to 19%, with fatty acid amide hydrolase 2 (FAAH2) showing the strongest effect: odds ratio (OR) = 1.19 (95% CI: 1.12, 1.25, q ≤ 0.01; **Table 1**).

### Proteins associated with a decreased risk of hospitalization as a result of COVID-19

After multiple testing correction ($p_{FDR}$ = 0.05), using genetic instruments and under the assumptions of Mendelian randomization, our results were consistent with nine blood markers being significantly causally associated with a decreased risk of hospitalization as a result of COVID-19 (**Figs 1 and S2**); the reverse associations with these nine blood markers were nonsignificant (**S3 Table**). Per SD increase in the respective blood marker the decreases in odds for hospitalization ranged from 7 to 20%, with the platelet endothelial cell adhesion molecule (PECAM-1) showing the strongest effect: OR = 0.80 (95% CI: 0.73, 0.87, q ≤ 0.01; **Table 1**).

### Hospitalization as a result of COVID-19 associated with protein levels

In addition, using genetic instruments and under the assumptions of MR, our results were consistent with hospitalization being significantly causally associated with decreased levels of macrophage inflammatory protein (MIP1b): beta = -0.16 (SE = 0.03), q ≤ 0.01; **Table 1**).

### Proteins associated with an elevated risk of need for respiratory support/ death due to COVID-19

After multiple testing correction ($p_{FDR}$ = 0.05), using genetic instruments and under the assumptions of Mendelian randomization, our results were consistent with five blood markers being causally associated with need for respiratory support/death due to COVID-19 (**Figs 2 and S3**); the reverse associations with these blood markers as outcomes were nonsignificant (**S3 Table**). Per standard deviation (SD) increase in these respective blood marker the increase in odds for respiratory support/death ranged from 12 to 35%, with glucosaminyl (N-Acetyl) transferase 4 (GCNT4) showing the strongest effect: OR = 1.35 (95% CI: 1.26, 1.44, q ≤ 0.01; **Table 1**).

### Proteins associated with a decreased risk of need for respiratory support/ death due to COVID-19

After multiple testing correction ($p_{FDR}$ = 0.05), using genetic instruments and under the assumptions of Mendelian randomization, our results were consistent with eight blood markers being causally associated with a statistically significantly decreased risk of need for respiratory support/death due to COVID-19 (**Figs 2 and S4**); the reverse associations with these blood markers were nonsignificant (**S3 Table**). Per standard deviation (SD) increase in the respective blood marker the increases in odds for respiratory support/death ranged from 11 to 27%, with the platelet endothelial cell adhesion molecule (PECAM-1) showing the strongest effect size: OR = 0.73 (95% CI: 0.63, 0.83, q ≤ 0.01; **Table 1**).

### Need for respiratory support/death due to COVID-19 associated with protein levels

In addition, using genetic instruments and under the assumptions of Mendelian randomization, our results were consistent with respiratory support/death due to COVID-19 being

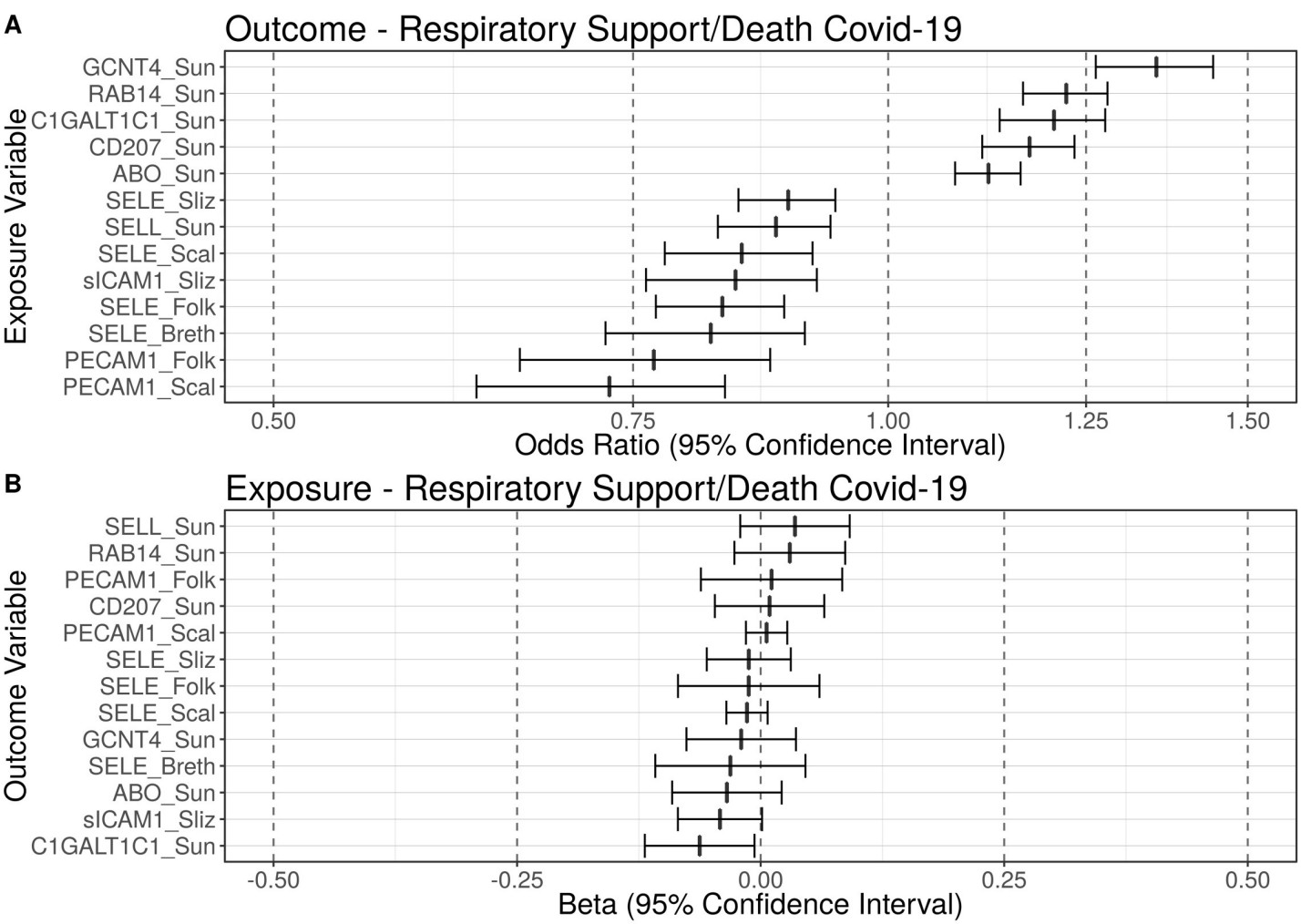

**Fig 2. Blood markers putatively causally associated with need for respiratory support/death due to COVID-19.** Summary figure of the false discovery rate-corrected ($p_{FDR} = 0.05$) Mendelian randomization results using the Generalised Summary data-based Mendelian randomization (GSMR) method. Using genetic instruments and under the assumptions of Mendelian randomization, this figure displays: (A) Summary figure when respiratory support/death-COVID-19 is the outcome of interest; (B) Summary figure when respiratory support/death-COVID-19 is the exposure of interest. Odds ratios (ORs) of blood markers causally associated with the need for respiratory support/death due to COVID-19 are displayed on the x-axis (with 95% confidence intervals). The blood markers are displayed on the y-axis. The dashed line at one represents an odds ratio of one (i.e., no effect). Using genetic instruments and under the assumptions of Mendelian randomization, five blood markers were causally associated with a significantly increased risk for need for respiratory support/death due to COVID-19 and eight blood markers were causally associated with a significantly decreased risk for respiratory support/death ($q_{FDR} \leq 0.05$). ABO = ABO system transferase; C1GALT1C1 = C1GALT1 specific chaperone 1; CD207 = langerin; GCNT4 = glucosaminyl (N-Acetyl) transferase 4; LCTL = lactase-like protein; PECAM1 = platelet endothelial cell adhesion molecule; RAB14 = ras-related protein rab-14; SELE = E-selectin; SELL = L-selectin; sICAM1 = soluble intercellular adhesion molecule-1.

significantly causally associated with decreased levels of neprilysin (NEP): beta = -0.28 (SE = 0.07, q ≤ 0.01; **Table 1**).

## Sensitivity analyses

**Additional Mendelian randomization analyses.** To further increase confidence in the findings, we additionally filtered our results for those concurrent with results calculated by additional Mendelian randomization methods (Maximum likelihood, MR Egger, Simple median, Weighted median, Inverse variance weighted (IVW), IVW radial, IVW multiplicative random effects, IVW fixed effects, Simple mode, Weighted mode).

Our sensitivity analyses confirmed the robustness of the association between 14 out of 15 blood markers and hospitalized-COVID-19, with ATP2A3 not satisfying the sensitivity analysis criteria (see S4 and S5 Tables, for a full breakdown of these sensitivity analyses).

Our sensitivity analyses also confirmed the association between GCNT4_Sun, RAB14_Sun, CD207_Sun, SELL_Sun, sICAM1_Sliz, SELE_Folk, PECAM1_Scal and severe-COVID-19. C1GALT1C1_Sun, ABO_Sun, sEselectin_Sliz, SELE_Scal, SELE_Breth, PECAM1_Folk did not survive our sensitivity analyses (see S4 and S5 Tables for a full breakdown of these sensitivity analyses).

**Strength of genetic instruments.** Given that we are using a reduced p-value threshold to identify SNPs as our genetic instruments, we calculated F-statistic and I-squared on all our significantly associated proteins. We found that with the exception of ATP2A3, all F-statistics were larger than 20 and all I-squared statistics were larger than 0.9, suggesting strong genetic instruments for use in analyses [44–46]. See S2 Table for full results of these analyses.

**Linkage disequilibrium analyses.** Given that many inflammatory and immunomodulatory proteins share genetic loci which may therefore be driving associations via SNPs in high linkage disequilibrium, we computed pairwise linkage disequilibrium statistics for all SNPs used as instrumental variables for blood markers significantly associated with our outcomes.

In our analysis of hospitalization as a result of COVID-19, we used 296 unique SNPs as instrument variables for blood proteins, in 30 cases, one SNP was used as an instrument for two or more different proteins, 27 of which were located on chromosome 9. In the assessment of pairwise LD, 40 SNP pairs (based on 55 unique SNPs) showed high LD ($r^2 > 0.6$). Most pairs ($n = 37$) were located on chromosome 9 carrying the ABO gene (S6 Table and S5 Fig).

In our analysis of the need for respiratory support/death due to COVID-19, we used 305 unique SNPs as instrument variables for blood proteins; however, in 31 cases, one SNP was used twice as an instrument for two or more different proteins, 28 of which were located on chromosome 9. Furthermore, assessing pairwise LD between all SNPs, we found 50 SNP pairs (based on 65 unique SNPs) in high LD ($r^2 > 0.6$). Most pairs ($n = 44$) were located on chromosome 9 carrying the ABO gene (S7 Table and S6 Fig) [47–49].

**cis-SNP effects from significantly associated proteins.** In order to establish whether the significant associations identified in our study were being driven by cis-regulatory variants, we identified cis-SNPs from all significant blood markers and performed Mendelian randomization analyses using only these SNPs with the respective COVID-19 GWASs. Out of the 28 exposure-outcome pairs, only seven (25%) where based on at least one cis-SNP and could therefore be analysed. The results show that ABO_Sun and SFTPD_Breth cis-SNPs are significantly associated with hospitalization, and ABO_Sun and sICAM-1_Sliz cis-SNPs are significantly associated with need for respiratory support/death. All other significant associations are deemed to be driven by trans-SNPs (S8 Table).

**Mendelian randomization analyses with body mass index.** High BMI has been robustly associated with both inflammatory cytokine levels in blood and an increased risk of severe COVID-19 [9,40,41]. After validating this relationship by performing Mendelian randomization analysis between BMI and the two COVID-19 outcomes (S9 Table), we performed bidirectional Mendelian randomization analyses between all significant blood markers and BMI, using the largest publicly available BMI GWAS [50]. Our results indicate that genetic susceptibility for higher BMI is associated with higher levels of SELE_Scal, C1GALT1C1_Sun, SELE_-Folk, KEL_Sun, SELL_Sun, RAB14_Sun and SFTPD_Breth. In addition, the genetic susceptibility for higher levels of LCTL_Sun, SELE_Sliz, SFTPD_Breth, PECAM-1_Scal and RAB14_Sun is associated with higher BMI (see S10 Table for full results). Note that some protein GWASs had controlled for BMI, whereas others, the majority of which show a significant association with BMI, did not (see S1 Table).

**Table 2. Groupings of statistically significantly associated proteins by their biological processes.**

| Protein | Association |
|---------|-------------|
| **Blood group proteins** | |
| ABO | Risk of hospitalization & need of respiratory support or death due to COVID |
| KEL | Protection against hospitalization |
| **Antigen recognition** | |
| CD207 | Risk of hospitalization & need of respiratory support or death due to COVID |
| SFTPD | Protection against hospitalization |
| **Adhesion molecules** | |
| SELL | Protection against hospitalization & need of respiratory support or death due to COVID |
| SELE | Protection against hospitalization & need of respiratory support or death due to COVID |
| PECAM-1 | Protection against hospitalization & need of respiratory support or death due to COVID |
| sICAM-1 | Protection against need of respiratory support or death due to COVID |
| **Transporters** | |
| RAB14 | Risk of hospitalization & need of respiratory support or death due to COVID |
| ATP2A3 | Protection against hospitalization |
| **Enzymes** | |
| GCNT4 | Risk of hospitalization & need of respiratory support or death due to COVID |
| C1GALT1C1 | Risk of hospitalization & need of respiratory support or death due to COVID |
| FAAH2 | Risk of hospitalization |
| LCTL | Protection against hospitalization |

Note: ABO = ABO system transferase; ATP2A3 = ATPase sarcoplasmic/endoplasmic reticulum Ca2+ transporting 3; C1GALT1C1 = C1GALT1 specific chaperone 1; CD207 = langerin; FAAH2 = fatty acid amide hydrolase 2; GCNT4 = glucosaminyl (N-Acetyl) transferase 4; KEL = Kell metallo-endopeptidase (Kell Blood Group); LCTL = lactase-like protein; PECAM1 = platelet endothelial cell adhesion molecule; RAB14 = ras-related protein rab-14; SELE = E-selectin; SELL = L-selectin; SFTPD = surfactant protein D; sICAM1 = soluble intercellular adhesion molecule-1.

**Pathway analysis.** Proteins significantly associated with respiratory support/death as a result of COVID-19 were significantly enriched in three KEGG pathways: "Cell adhesion molecules" (hsa04514), "Mucin type O-glycan biosynthesis" (hsa00512) and "Malaria" (hsa05144). No enrichment was found at the defined significance threshold for hospitalization as a result of COVID-19, however "cell adhesion molecules" and "Mucin type O-glycan biosynthesis" (hsa00512) showed the strongest signal ($q_{FDR} = 0.07$)

## Discussion

Using genetic instruments and under the assumptions of Mendelian randomization, our proteome-wide analyses are consistent with higher levels of certain blood proteins being causally associated with risk of being hospitalized due to COVID-19, and subsequently experiencing the most severe form including respiratory support or ending lethal (i.e., respiratory support/death in the following). All these proteins have detectable blood plasma or serum levels. For our discussion, we grouped the proteins by function in **Table 2** and provided more detail in **S11 Table**.

It is important to note that, in our analyses, we did not identify typical canonical immune proteins, such as interleukin 6 or C-reactive protein [51,52]. This suggests that with a larger database of proteins we can pinpoint non-canonical immunomodulatory proteins relevant to disease pathophysiology. We did, however, estimate associations for some proteins twice, as they were measured separately in independent GWASs. Results from these analyses displayed

the same direction of effects, rendering them a direct replication and increasing the validity of our findings.

## Blood group proteins

In the blood group protein group, using genetic instruments and under the assumptions of Mendelian randomization, our findings were consistent with ABO being causally associated with both an increased risk of hospitalization as well as the requirement of respiratory support or death by COVID-19 (i.e., respiratory support/death). ABO is an enzyme with glycosyltransferase activity that determines the ABO blood group of an individual [53]. However, the precise blood group associated with the increased risk for hospitalization as a result of COVID-19 cannot be determined from our results, as the probe for the blood marker measures both the A and B isoform of the protein while not showing a signal for O. Given the underlying British population of the original GWAS, A should be the more prevalent blood group (24%) in the sample compared to B (8%) [54]. Nevertheless, it is more likely that A, B, or the combination of A and B is associated with higher risk for hospitalization. Our findings confirm previous reports of the ABO blood group system being an important risk factor for a severe COVID-19 infection. For example, the proportion of group A is higher in COVID-19 positive individuals than in controls [55–60], and group A has been associated with higher mortality [61]. All evidence taken together suggests that blood group A is the more likely candidate for follow-up studies. Additionally, KEL, which is part of the complex Kell blood group system that contains many highly immunogenic antigens [62], was associated with a decreased risk of hospitalization as a result of COVID-19. This supports the notion that Kell negative individuals may be more susceptible to COVID-19 [63].

## Antigen recognition

In our study, CD207, also known as langerin, was associated with hospitalization as well as the requirement of respiratory support or death by COVID-19. This protein is exclusively expressed in Langerhans cells (LC)–the first dendritic cells to encounter pathogens entering the body via the mucosa or skin [64]. Langerin binds COVID-19 glycoprotein glycans; however, it does not mediate transfection of COVID-19 pseudovirions in a T-lymphocyte cell line [65], rendering its role in COVID-19 infections inconclusive [66]. Our findings showed evidence consistent with high levels of SFTPD potentially protecting against COVID-19 hospitalization. SFTPD is strongly expressed in lung, brain, and adipose tissue, and contributes to the lung's defense against microorganisms, antigens, and toxins [67]. SFTPD also interacts with COVID-19 spike proteins [68]. In COVID-19, expression findings are mixed: some studies show that SFTPD is highly expressed in the lungs of COVID-19 patients [68], whereas others evidence a decreased expression [69]. Moreover, another study which investigated gene expression patterns in COVID-19-affected lung tissue and SARS-CoV-2 infected cell-lines, report a downregulation of SFTPD along with several regulatory partners [70]. Given its role in immunomodulation and air exchange in the lung, this supports our finding that higher levels of SFTPD may be causally associated with COVID-19 immunity [71,72]. Although more research is needed; however, ours and others' findings imply that SFTPD may protect against severe forms of COVID-19.

## Adhesion molecules

Using genetic instruments and under the assumptions of Mendelian randomization, our analysis was consistent with the adhesion molecules SELE, SELL, and PECAM-1 being causally associated with a decreased risk of both hospitalization and the requirement of respiratory

support/death by COVID-19, while ICAM-1 was only protective against respiratory support/ death. This is in keeping with results from out pathway analyses which suggest a significant enrichment in the KEGG pathway "cell adhesion molecules". Studies have suggested that late stage COVID-19 is an endothelial disease [73]. The vascular endothelium is the crucial interface between blood and other tissues, regulating vascular structure, permeability, vasomotion, inflammation, and oxidative stress [73]. SELL and SELE are members of the selectin class of leukocyte adhesion molecules, which facilitate slow rolling of blood leukocytes along the endothelium [73]. Specifically, SELL promotes initial tethering and rolling of leukocytes to the endothelium [74,75] and SELE is responsible for the accumulation of blood leukocytes at sites of inflammation by mediating the adhesion of cells to the vascular lining [76,77]. The firm binding of leukocytes to the endothelial surface depends upon other molecules, such as PECAM-1, which is a cell adhesion molecule required for leukocyte transendothelial migration under most inflammatory conditions [78,79], and our results were consistent with it being protective against hospitalization. Once tightly bound, chemoattractant cytokines can signal to the bound leukocytes to traverse the endothelial monolayer and enter tissues where they can combat pathogenic invaders and initiate tissue repair [80]. This may be one of the biological explanations why we saw elevated levels of SELL, SELE, and PECAM-1 as being protective against hospitalization. ICAM-1–our results being consistent with it being protective against hospitalization–mediates cell-cell adhesion and is involved in inflammation [81]. Contrary to our findings, higher ICAM-1 levels have been associated with COVID-19 severity [82,83], requiring follow-up investigations. In summary, molecules that mediate the interaction between immune cells and blood vessels may be important in late stage COVID-19 and moderate severity.

## Transporter molecules

In the protein transporter/trafficking group, using genetic instruments and under the assumptions of Mendelian randomization, our results were consistent with RAB14 being causally associated with an increased risk of hospitalization and respiratory support/death, whereas ATP2A3 may protect against hospitalization. Rab proteins are central regulators of phagosome maturation; RAB14 particularly regulates the interaction of phagosomes with early endocytic compartments [84]. One study identified RAB14 GTPases as a critical COVID-19 host factor: coronaviruses hijack Rab GTPase in host cells to replicate [85]. Additionally, whole genome analysis of COVID-19 lung tissue identified RAB14 polymorphisms that alter its binding to some miRNAs [86]. Therefore, Rab GTPases could be therapeutic targets. ATP2A3 is a magnesium-dependent ATP hydrolase, transports calcium from the cytosol into the sarcoplasmic/ endoplasmic reticulum involved in muscular excitation/contraction and contributes to calcium sequestration [87,88]. Note that ATP2A3 has previously been genetically associated with severe COVID-19, but its exact role in infection remains unclear [89]. Cardiac failure in severe COVID-19 has been reported [90,91], hence, ATP2A3 may be involved as it regulates cardiomyocyte contraction [92]. However, note that ATP2A3 did not survive our sensitivity analyses, so this finding would need further validation.

## Enzymes

In the enzyme group, which consists predominantly of glycosylases and hydrolases, using genetic instruments and under the assumptions of Mendelian randomization, GCNT3, a member of the GCNT family, was consistent with increasing the risk for hospitalization and respiratory support/death. GCTN3 proteins mediate mucin synthesis, branching, and oligomerization [93], a pathway showing a significant enrichment in our analyses. Glycosylation of COVID-19 viral surface antigens may help the virus evade the host immune system by

shielding its protein surface and, therefore, may prevent the development of an effective immune response [94]. In addition, as part of our innate immunity, the epithelial barrier made up of mucins acts as the first line of defense [95]. Our analysis also showed evidence consistent with C1GALT1C1 being risk increasing. C1GALT1C1 is a molecular chaperone required for the expression of active T-synthase, the only enzyme that glycosylates the Tn antigen [96]. Anti-Tn antibodies are lower in COVID-19 patients than noninfected individuals and individuals with blood group A [97]. Anti-Tn antibodies may protect against COVID-19 [97]. The glycosylation of Tn by C1GALT1C1 may suppress these antibodies. The glycobiology of COVID-19 includes glycans on viral proteins and host glycosaminoglycans that are critical in infections [98]. FDA-approved drugs, such as glycans for vaccines, anti-glycan antibodies, recombinant lectins, lectin inhibitors, glycosidase inhibitors, polysaccharides, and numerous glycosides may be repurposing targets for COVID-19 [98]. Our analysis also showed evidence consistent with FAAH2 being risk increasing for hospitalization as a result of COVID-19. FAAH2, is a fatty acid hydrolase involved in endocannabinoid uptake and inactivation [99]. Cannabinoids may reduce pulmonary inflammation through immunomodulation, decrease polymorphonuclear leukocytes infiltration, reduce fibrosis, decrease viral replication, and modulate the 'cytokine storm' in COVID-19 [100–103]. Cannabinoids have been suggested as anti-inflammatory treatment in COVID-19 [102,103]. Our results are also consistent with LCTL, being protective against hospitalization. LCTL is a glycosidase which hydrolyses glycosidic bonds. Little is known about this protein, and nothing in the context of COVID-19.

Using genetic instruments and under the assumptions of Mendelian randomization, our analysis was also consistent with hospitalization due to COVID-19 decreasing levels MIP1b. MIP1b is a major proinflammatory factor, acting as a chemoattractant for natural killer cells [104]. This association has been reported previously [105], with studies suggesting that MIP1b is a key mediator in the immune response against COVID-19 [106]. Also, in line with our findings of respiratory support/death due to COVID-19 decreasing levels of NEP, NEP protects against pulmonary inflammation and fibrosis [107]. Other studies suggest repurposing of roflumilast, a treatment for chronic obstructive pulmonary disorder, that increases NEP activity and, hence, increases anti-inflammatory activity [107].

Our study has several limitations. First, although we required confirmation of our findings by several Mendelian randomization methods, we set the $p$-value threshold for selecting genetic variants as our instruments at genome-wide suggestive significance ($p < 5 \times 10^{-6}$) to identify enough SNP instruments for each protein to run Mendelian randomization analyses. Some genetic variants may therefore be false positive associations with protein levels. This procedure is common in Mendelian randomization analyses [27–31]. Additionally, we report F-statistics and I-squared statistics per SNP instrument (**S2 Table**) so that the quality of our instruments is transparent. This step was necessary as the available GWASs of the blood proteins are still of limited sample size, and therefore statistical power. However, we identified several blood markers robustly associated with COVID-19, including ABO, suggesting that our SNP instruments pick up true associations and potential underlying biology of COVID-19. In addition, we note that for most associations (such as between FAAH2 and hospitalized COVID-19, or sICAM1 and severe COVID-19) we observe a clear linear relationship (**S1–S4 Figs**). However, some associations, such as between ABO and hospitalized COVID-19, or GCNT4 and severe COVID-19, display SNP effects between the exposure and outcome with very large standard errors. While these SNPs have less statistical power, the joint effect over all SNPs remains significant. Second, some blood marker GWASs were excluded from our analyses due to unavailability, therefore, we may have missed associations with these markers. Third, although the COVID-19 GWASs used in our analyses were carefully chosen to

represent two different phenotypes–hospitalization and respiratory support/death due to COVID-19–other GWAS may be better powered for identifying severe COVID-19 pheno-types. However, those were not publicly available when we conducted our study. Fourth, some of our SNP instruments for blood proteins are either the same or are in high LD, potentially tagging the same causal variant. Although this may indicate pleiotropy across blood markers, these instruments only represent a minority of our instruments (28% for hospitalization and 31% for respiratory support/death, respectively). Furthermore, a significant proportion of these SNPS are either used as instruments for the same protein from separate studies (e.g., in the case of SELE or PECAM), or are used as pleiotropic instruments for the different adhesion molecules (the family of SELE, PECAM-1, ICAM and SELL). Therefore, they should not overly influence our results. Fifth, the proteins in the original GWAS were measured in blood, thus not necessarily reflecting their intracellular concentrations. Therefore, it is difficult to draw conclusions on intracellular concentrations based on our results and further cellular research is required. Sixth, we used several robust MR methods with varying abilities to detect heteroge-neity and pleiotropy; however, residual heterogeneity or pleiotropy may still be present. How-ever, this is common to most Mendelian randomization analyses. Sixth, our sensitivity analyses demonstrate that a genetic predisposition to high BMI is significantly associated with some of our blood markers. High BMI is also associated with severe COVID-19 ([41], **S9 Table**), suggesting that BMI may drive risk of severe forms of COVID-19 and the change in blood protein levels, potentially confounding our results. However, our Mendelian randomiza-tion analyses showed that higher BMI is causal for higher levels of the COVID-19 protective proteins SELE, KEL, SELL, and causal for lower levels of the COVID-19 risk increasing pro-teins C1GALT and RAB14. Our sensitivity analysis therefore shows that BMI influences these blood markers in opposite directions than would be concurrent with confounding effects. This indicates the effects of these proteins are independent of BMI. Also, some blood protein GWASs controlled for BMI which should in the first place reduce the potential confounding (**S1 Table**). Finally, our findings are based on measurements in ancestral European popula-tions due to data availability; therefore, future endeavors should include participants from more diverse ancestry.

Our results highlight the utility of applying large scale Mendelian randomization analyses to identify blood markers that may be causal for severe COVID-19. Using genetic instruments and under the assumptions of Mendelian randomization, our findings are consistent with higher levels of GCNT4, RAB14, C1GALT1C1, CD207 and ABO causally increasing the risk of both hospitalization and need of respiratory support or death due to COVID-19, and higher levels of FAAH2 increasing the risk of hospitalization. Our results were also consistent with higher levels of a number of adhesion molecules, including SELE, SELL, PECAM-1 and ICAM-1, as being protective against both hospitalization and a need of respiratory support or death. This adds to a growing body of evidence for the involvement of adhesion and endothe-lial dysfunction in severe COVID-19. Moreover, our results were consistent with higher levels of LCTL, SFTPD and KEL being protective against hospitalization alone. Together, our find-ings support previous findings and identify novel blood markers associated with a severe COVID-19 phenotype, indicating possible avenues to develop prognostic biomarkers and therapeutics for COVID-19.

## Supporting information

**S1 Table. A breakdown of all studies from which inflammatory marker genome-wide asso-ciation study (GWAS) data originated.**
(DOCX)

**S2 Table. Validation of our genetic instruments.**
(DOCX)

**S3 Table. Reverse effect results from Mendelian randomization analyses.**
(DOCX)

**S4 Table. Results from sensitivity analyses for all markers and risk hospitalization as a result of COVID-19.**
(DOCX)

**S5 Table. Results from sensitivity analyses for all markers and respiratory support/death as a result of COVID-19.**
(DOCX)

**S6 Table. Table indicating the heterogeneous SNP used as instruments for at least 2 bio-markers identified in the hospitalization from COVID-19 GWAS.**
(DOCX)

**S7 Table. Table indicating the heterogeneous SNP used as instruments for at least 2 bio-markers identified in the respiratory support/death as a result of COVID-19 GWAS.**
(DOCX)

**S8 Table. cis-SNP effects from the significantly associated proteins.**
(DOCX)

**S9 Table. COVID-19 associations with BMI.**
(DOCX)

**S10 Table. Blood marker associations with BMI.**
(DOCX)

**S11 Table. Details on the tissue, function, and Covid-19 relevance of each significant blood biomarker.**
(DOCX)

**S1 Fig. Effect plots of blood markers associated with an increase in hospitalized COVID-19 risk.**
(DOCX)

**S2 Fig. Effect plots of blood markers associated with a decrease in hospitalized COVID-19 risk.**
(DOCX)

**S3 Fig. Effect plots of blood markers associated with an increase in risk of respiratory support/death as a result of COVID-19.**
(DOCX)

**S4 Fig. Effect plots of blood markers associated with a decrease in risk of respiratory support/death as a result of COVID-19.**
(DOCX)

**S5 Fig. Plot indicating the number of heterogeneous SNP instruments on chromosome 9 from biomarkers identified in the hospitalization from COVID-19 GWAS.**
(DOCX)

**S6 Fig. Plot indicating the number of heterogeneous SNP instruments on chromosome 9 from biomarkers identified in the respiratory support/death as a result of COVID-19**

GWAS.
(DOCX)

**S1 Data. The sheets in this spreadsheet displays all of our results, including a summary table of our main findings, all sensitivity analyses (different MR methods, sensitivity analyses relating to cis-SNP effects of all blood markers, sensitivity analyses relating to the association between significant blood markers and BMI, sensitivity analyses relating to the genetic instruments used in our analyses), the lists of the associations between both COVID phenotypes and all blood markers, in both directions, as well the full details of all SNP instruments used for the association with significant blood markers.**
(XLSX)

# Acknowledgments

The views expressed are those of the author(s) and not necessarily those of the National Health Service (NHS), the National Institute of Health Research (NIHR), King's College London, or the Department of Health and Social Care.

# Author Contributions

**Conceptualization:** Alish B. Palmos, Vincent Millischer, David K. Menon, Timothy R. Nicholson, Leonie S. Taams, Benedict Michael, Christopher Hübel, Gerome Breen.

**Data curation:** Alish B. Palmos, Vincent Millischer, Christopher Hübel.

**Formal analysis:** Alish B. Palmos, Vincent Millischer, Christopher Hübel.

**Funding acquisition:** Benedict Michael, Gerome Breen.

**Investigation:** Alish B. Palmos, Vincent Millischer, Christopher Hübel.

**Methodology:** Alish B. Palmos, Vincent Millischer, Geraint Sunderland, Christopher Hübel.

**Project administration:** Alish B. Palmos, Vincent Millischer, Christopher Hübel.

**Resources:** Gerome Breen.

**Software:** Alish B. Palmos, Vincent Millischer.

**Supervision:** Michael J. Griffiths, Christopher Hübel, Gerome Breen.

**Validation:** Alish B. Palmos, Vincent Millischer, Christopher Hübel.

**Visualization:** Alish B. Palmos, Vincent Millischer.

**Writing – original draft:** Alish B. Palmos, Vincent Millischer, Christopher Hübel, Gerome Breen.

**Writing – review & editing:** Alish B. Palmos, Vincent Millischer, David K. Menon, Timothy R. Nicholson, Leonie S. Taams, Benedict Michael, Geraint Sunderland, Michael J. Griffiths, Christopher Hübel, Gerome Breen.

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
