## [Decision Letter · Decision Letter 0]

13 Sep 2021

Dear Dr Millischer,

Thank you very much for submitting your Research Article entitled 'Proteome-wide Mendelian randomization identifies causal links between blood proteins and severe COVID-19' to PLOS Genetics.

The manuscript was fully evaluated at the editorial level and by independent peer reviewers. The reviewers appreciated the attention to an important problem, but raised some substantial concerns about the current manuscript. Based on the reviews, we will not be able to accept this version of the manuscript, but we would be willing to review a much-revised version. We cannot, of course, promise publication at that time.

If you decide to revise the manuscript for further consideration at PLOS Genetics, please aim to resubmit within the next 60 days, unless it will take extra time to address the concerns of the reviewers, in which case we would appreciate an expected resubmission date by email to plosgenetics@plos.org.

[LINK]

We are sorry that we cannot be more positive about your manuscript at this stage. Please do not hesitate to contact us if you have any concerns or questions.

Yours sincerely,

Chris Cotsapas, PhD

Associate Editor

PLOS Genetics

Scott Williams

Section Editor: Natural Variation

PLOS Genetics

Reviewer's Responses to Questions

**Comments to the Authors:**

Reviewer #1: In this manuscript, Palmos, Millischer et al. describe a mendelian randomization study between ~4000 blood protein levels and a sever COVID-19 (incidence vs. population controls). Obviously this is a pressing are of research, although insights from host genetic analyses have arguably been limited to date.

Overall, the manuscript follows standard practices in the MR literature, the presentation was clear, and the results might be important. However, the mendelian randomization approach has serious limitations, and my comments are focused on how the authors might engage with those limitations. I don’t think it is necessary for every one of the suggested analyses to unambiguously support a causal relationship.

The most important limitation of MR is that pleiotropy, and in particular genetic correlations, lead to false positive “causal” relationships. This is extremely common, even with commonly-used sensitivity analyses (e.g. MR-Egger) (O’Connor and Price 2018, Verbanck et al 2018). Two recent methods are more robust (LCV and CAUSE, Morrison et al 2020), but I expect that both will be underpowered for these phenotypes. It is encouraging that the authors tested for a reverse causal effect and did not find one, but this is not unexpected given the phenotypes involved (and limited power in the COVID study). I have three suggestions to determine the level of confidence in the putative causal relationships:

1. Please show scatter plots for the effect size estimates of your instrumental variables on the protein level and on COVID risk, respectively. These are helpful to evaluate whether the correlation is uniform, with all variants affecting protein level also affecting risk, or whether it is driven by just one or two shared variants.

2. Please report the effect, if any, of variants that are in the cis region of the protein that they encode. Such variants make relatively strong instrumental variables, because their effects are unlikely to be mediate by a confounder in trans.

3. Please check if any of your instrumental variables are pleiotropically associated with BMI, which would be an obvious confounder. (It may affect both protein levels and severe COVID risk). Also, please check which of your results change if the HLA region (which has very long-range LD and unusually strong associations with numerous phenotypes) is excluded from the analysis.

I don’t think it is necessary that these pieces of evidence provide unambiguous evidence of a causal relationship for all, or even any, of the inferred relationships; it is appropriate to publish equivocal evidence when it is the best that is available, if it is communicated accurately.

I’ll just highlight one particular result as needing more attention and discussion: the ABO finding seems concordant with early reports of a blood-type association, but my impression was that recent evidence has shown this to be a false positive, possibly due to stratification. Is the ABO association supported by any variants outside of the disputed locus?

Reviewer #2: The authors seek to understand if protein levels are causally associated to severe COVID19 infection. They use the Mendelian Randomisation framework in which SNPs associated with protein levels are instrument variable used to test for the causal relationship between proteins as exposure traits and COVID-19 hospitalization as outcome. This is an interesting and potentially important question as such proteins could be targeted by drugs. The MR framework is a very cost-effective approach to address this question.

The analyses use published methods and results from published studies. The analyses are likely to be repeated over time as GWAS sample sizes become larger, but it makes sense to conduct the analyses now if suitable SNP instruments exist.

To be useful to the research community the study needs to clearly build on other studies and be reproducible.

Major comments

1. Poor choice of COVID GWAS

a) The primary phenotype is taken from the host genetic initiative release 5, which makes sense as these were presented in the Nature paper (reference 7) published earlier this year. From this paper, the authors have selected the GWAS results: “A2_ALL_leave_UKBB: Very severe respiratory confirmed covid (5,870) vs. population (1,155,203)” incorrectly calling it the European only sample. I agree it makes sense to use the European sample, they should have selected: “A2_ALL_eur: Very severe respiratory confirmed covid (5,101) vs. population (1,383,241)”

b) The authors have chosen to label this phenotype “hospitalized-COVID-19 GWAS” even though Ganna et al called this phenotype “critically ill COVID-19+” ; Ganna et al used the name “Hospitalised COVID” for a less severe phenotype 13K case 2M controls (or just Europeans B2_ALL_eur: (9,986) vs. population (1,877,672)). I don’t think it is helpful to mix up names in this way. No doubt the reason for this was because the authors tried to select a what they thought was a more phenotype selecting from release 4, “Very severe respiratory confirmed covid vs. not hospitalized covid” 269 cases vs 688 controls (A1_ALL). It is not clear why this sample was chosen it was very small, has no GWS SNPs, and while I don’t think this analysis is needed (the very large population control makes more sense) release 5 has much bigger samples for this phenotype 4,829 and 11,816 (B1_ALL_eur).

Summary: Use only A2_ALL_eur release 5 (or 6 if now available).

2. Non-reproducibility of protein GWAS.

From what is written or available in the supplement I am not sure if others could repeat the analysis.

a) Supplementary Table 1 lists 10 studies but five of these use the OLINK panel – were these meta-analysed?

b) “We tested 4,366 associations with hospitalized-COVID-19 as the exposure and the blood proteins as outcome”. So is 4,366 the number of proteins or the number of SNPs? Suggest you provide a supplementary data file listing the SNPs used as instruments for each protein.

Minor comments:

1) Referring to the differences in GWAS results severe vs hospitalised a statement is made “The findings suggest that one set of individual genetic variants may be responsible for hospitalization, and once hospitalized, another set of genetic variants may be responsible for respiratory failure and death.” This statement and elsewhere provides a feeling that the researchers have not fully engaged with the severe-COVID19 phenotype. This statement makes these findings sound novel, but this difference is well known clinically. For example, the GENOMICC GWAS (Pairo-Castineira et al) said “In the UK, the group of patients admitted to critical care is relatively homogeneous, with profound hypoxaemic respiratory failure being the archetypal presentation “ and “Patients admitted to intensive care units in the UK during the first wave of COVID-19 were, on average, younger and less burdened by comorbid illnesses than the hospitalized population” – citing Docherty et al.

2) What are the units of the Beta and odds ratios. Are the units the same for all analyses?

3) gwasrapidd not gwasrapid

4) Statements such as “the genetic liability for exposure X is causal for outcome Y” are made on several occasions. Tthese statements are commonly found in MR publications but in my opinion miss the point about MR. The point is that using genetic instruments and under the assumptions of MR results are consistent with exposure X being causal for outcome Y. There is a subtle difference in meaning here

5) In this study, the 1,000 Genomes dataset was the reference dataset (18). Do you mean the European ancestry samples to match the EUR ancestry GWAS?

6) To get sufficient SNP instruments to conduct MR, the significance threshold was set at p < 5 x 10-6. This decision is justified through reference to 5 papers Ref 26-30, which includes one paper by the authors, and three rather old papers. Just because other papers have passed through peer review it does not justify the approach, this must be justified by a methodological paper. Generally, I believe use of weak instruments is not supported.

7) Given that the incorrect GWAS was selected I haven’t focussed on the results.

**Have all data underlying the figures and results presented in the manuscript been provided?**

Reviewer #1: Yes

Reviewer #2: **No: **See major point 2. Suggest that a supplementary data file listing the SNPs used as instruments for each protein is provided.

PLOS authors have the option to publish the peer review history of their article (what does this mean?). If published, this will include your full peer review and any attached files.

Reviewer #1: **Yes: **Luke O'Connor

Reviewer #2: No

---

## [Decision Letter · Decision Letter 1]

17 Dec 2021

Dear Dr Millischer,

Thank you very much for submitting your Research Article entitled 'Proteome-wide Mendelian randomization identifies causal links between blood proteins and severe COVID-19' to PLOS Genetics.

The manuscript was fully evaluated at the editorial level and by independent peer reviewers. The reviewers appreciated your responses to their previous comments, but identified some minor remaining concerns that we ask you address in a revised manuscript. In particular, reviewer 1 had a handful of remaining comments. We also want to draw your attention to reviewer 2's initial minor comment 4, about the language of causality. There are some remaining references to causality of genetic liability of an exposure, rather than the exposure itself, in the abstract, and on pp 25, 31 etc. We ask that you correct these in your revisions.

We therefore ask you to modify the manuscript according to the review recommendations. Your revisions should address the specific points made by each reviewer.

[LINK]

Yours sincerely,

Chris Cotsapas, PhD

Associate Editor

PLOS Genetics

Scott Williams

Section Editor: Natural Variation

PLOS Genetics

Reviewer's Responses to Questions

**Comments to the Authors:**

Reviewer #1: The authors have performed analyses to address my comments, and I have only minor follow up comments remaining:

1. In Supplementary Figures 1-4, the authors show scatterplots for their significant associations. Most of these look quite good, better than a lot of MR analyses, indicating that most protein-associated SNPs have proportional effects on hospitalization risk. For example slCAM1, which also passed the cis-variant sensitivity test, looks like it is supported by a large number of instruments with consistent effects. In contrast the ABO plot looks fairly poor, with many ABO-associated SNPs having discordant effects on the outcome. I think these results merit some discussion in the main text. Also, please fix the figure labels in Supp Fig 1, some of which are cut off.

2. The BMI analyses do seem to indicate that confounding due to BMI is potentially quite important. Please add some discussion about whether the direction of effect that you observed was consistent with potential confounding – i.e., when BMI-associated variants are associated with increased protein levels, are those the same proteins for which increased protein-increasing variants are associated with increased COVID risk?

3. Regarding the cis analysis, you might report that the total number of proteins that had testable cis variants (which was smaller than I expected, if I understand correctly); initially I got the impression that most proteins failed this sensitivity analyses, when it was actually not applicable. Please add a caption to Supplementary Table 6, and cite it in the text as a Table not as a Note.

Reviewer #2: The authors have addressed my comments, except Point 4, I leave in the hands of the editor.

**Have all data underlying the figures and results presented in the manuscript been provided?**

Reviewer #1: Yes

Reviewer #2: Yes

PLOS authors have the option to publish the peer review history of their article (what does this mean?). If published, this will include your full peer review and any attached files.

Reviewer #1: **Yes: **Luke J O'Connor

Reviewer #2: No

---

## [Editor Report · Decision Letter 2]

18 Jan 2022

Dear Dr Millischer,

We are pleased to inform you that your manuscript entitled "Proteome-wide Mendelian randomization identifies causal links between blood proteins and severe COVID-19" has been editorially accepted for publication in PLOS Genetics. Congratulations!

Yours sincerely,

Chris Cotsapas, PhD

Associate Editor

PLOS Genetics

Scott Williams

Section Editor: Natural Variation

PLOS Genetics

Comments from the reviewers (if applicable):

**Data Deposition**

http://datadryad.org/submit?journalID=pgenetics&manu=PGENETICS-D-21-01126R2

**Press Queries**

---

## [Editor Report · Acceptance letter]

10 Feb 2022

PGENETICS-D-21-01126R2 

Proteome-wide Mendelian randomization identifies causal links between blood proteins and severe COVID-19 

Dear Dr Millischer, 

We are pleased to inform you that your manuscript entitled "Proteome-wide Mendelian randomization identifies causal links between blood proteins and severe COVID-19" has been formally accepted for publication in PLOS Genetics! Your manuscript is now with our production department and you will be notified of the publication date in due course.

With kind regards,

Zsofia Freund

PLOS Genetics

On behalf of:
